# Anti-Trypanosomatidae Activity of Essential Oils and Their Main Components from Selected Medicinal Plants

**DOI:** 10.3390/molecules28031467

**Published:** 2023-02-02

**Authors:** María Bailén, Cristina Illescas, Mónica Quijada, Rafael Alberto Martínez-Díaz, Eneko Ochoa, María Teresa Gómez-Muñoz, Juliana Navarro-Rocha, Azucena González-Coloma

**Affiliations:** 1Department of Preventive Medicine, Public Health and Microbiology, Faculty of Medicine, Universidad Autónoma de Madrid, 28049 Madrid, Spain; 2Research and Development Division, AleoVitro, 48160 Derio, Spain; 3Department of Animal Health, Faculty of Veterinary Sciences, University Complutense of Madrid, 28040 Madrid, Spain; 4Centro de Investigación y Tecnología Agroalimentaria de Aragón, Unidad de Recursos Forestales, 50059 Zaragoza, Spain; 5Instituto de Ciencias Agrarias, CSIC, 28006 Madrid, Spain

**Keywords:** natural products, *Phytomonas*, *Leishmania*, cytotoxicity, Lamiaceae, Asteraceae, thymol, carvacrol

## Abstract

Kinetoplastida is a group of flagellated protozoa characterized by the presence of a kinetoplast, a structure which is part of a large mitochondria and contains DNA. Parasites of this group include genera such as *Leishmania,* that cause disease in humans and animals, and *Phytomonas,* that are capable of infecting plants. Due to the lack of treatments, the low efficacy, or the high toxicity of the employed therapeutic agents there is a need to seek potential alternative treatments. In the present work, the antiparasitic activity on *Leishmania infantum* and *Phytomonas davidi* of 23 essential oils (EOs) from plants of the Lamiaceae and Asteraceae families, extracted by hydrodistillation (HD) at laboratory scale and steam distillation (SD) in a pilot plant, were evaluated. The chemical compositions of the EOs were determined by gas chromatography-mass spectrometry. Additionally, the cytotoxic activity on mammalian cells of the major components from the most active EOs was evaluated, and their anti-*Phytomonas* and anti-*Leishmania* effects analyzed. *L. infantum* was more sensitive to the EOs than *P. davidi*. The EOs with the best anti-kinetoplastid activity were *S. montana*, *T. vulgaris*, *M. suaveolens*, and *L. luisieri*. Steam distillation increased the linalyl acetate, β-caryophyllene, and trans-α-necrodyl acetate contents of the EOs, and decreased the amount of borneol and 1,8 cineol. The major active components of the EOs were tested, with thymol being the strongest anti-*Phytomonas* compound followed by carvacrol. Our study identified potential treatments against kinetoplastids.

## 1. Introduction

Kinetoplastida is a group of flagellated protozoa characterized by the presence of a particular structure, known as a kinetoplast, which is located in the single mitochondrion of the cell and is composed of mitochondrial DNA. Despite kinetoplastids having similarities in their genomic organization and cellular structures, they comprise a large variety of organisms, many of them pathogenic, and transmitted by different vectors, causing different diseases in multiple hosts [1]. Parasites of the genera *Leishmania* [2] and *Trypanosoma* [3] cause disease in humans, while pathogens of the genus *Phytomonas* are capable of infecting plants [4].

*Leishmania* spp. causes leishmaniasis, a disease affecting populations with low resources [5]. In spite of its great epidemiological importance, leishmaniasis is considered by WHO as a neglected disease [6]. Treatment is based on pharmacological therapies; however, many of the currently employed drugs present high toxicities, decreased efficacies, difficulties of administration, and high costs, and the emergence of resistant strains has been reported [7].

*Phytomonas* comprises a wide range of parasite species which infect a diversity of vascular plants, many of them of great economic importance, such as cashew, coffee, oil palm, tomato, orange, and grape, among others [8]. The main symptoms of the disease caused by these parasites are leaf chlorosis and root atrophy [9]. Currently, there are no effective treatments and the strategies for infection control are the felling of diseased plants or the removal of infected plant material [4]. Additionally, *Phytomonas* share similar structural components and similarities in life cycle with other kinetoplastids such as *Leishmania* or *Trypanosoma,* which allows them to be used a laboratory safe model in the search for new anti-kinetoplastids compounds [8].

Natural products are chemical compounds produced by living organisms that have been used in traditional medicine for thousands of years, still serving today as the basis for many pharmaceuticals [10]. They are environmentally friendly due to their biodegradability and low toxicity to mammals [11]. EOs are plant-derived natural products consisting of complex mixtures of volatile, lipophilic, low molecular weight compounds, with terpenoids and phenylpropanoids among their most common constituents [12], in which hydrocarbons and oxygenated compounds stand out [13]. The composition of EOs may be influenced by several factors, such as water availability, soil composition, climate, nutrients, organ used for extraction, pathogen attack, or type of extraction [14]. They play an important ecological role in plants by attracting pollinators and beneficial insects, protecting from heat and cold, and being used as chemical defenses against pests, pathogenic microorganisms, or vertebrate predators [15]. EOs have pharmacological interest because of their biocidal effects on pathogenic microorganisms, among other properties [16]. They present a broad pharmacological spectrum that includes analgesic, sedative, anti-inflammatory, antispasmodic, antimicrobial, antiprotozoal, and anthelmintic properties [16,17], as well as insecticidal effects [18], and can be used to overcome toxicity and the emergence of drug-resistant pathogens associated with routine treatments [14].

Moreover, promising effects have been observed from the combination of natural products and approved antileishmanial drugs. Meglumine antimoniate combined with capsaicin or piperine produced synergistic effects on *L. infantum* promastigotes and amastigotes, which can help to reduce the amount of antimonials administered and its associated toxicity [19]. Another example was the effect of an ethanolic extract of *Moringa oleifera* in combination with amphotericin B on *L. major* amastigotes and promastigotes [20].

Several studies have determined that EOs from plant species belonging to the genera *Artemisia* [21], *Thymus, Lavandula* [14], *Salvia,* and *Origanum* [22], among others [23], exhibit antiparasitic properties against multiple protozoa of the kinetoplastida class.

There is a need to seek potential alternative treatments against *Leishmania* sp. and *Phytomonas* sp., among other kinetoplastids, considering the low efficacy or high toxicity of some of the current treatments against Trypanosomatids. In the present work, the antiparasitic activity of 23 EOs from plants of the Lamiaceae (*Salvia*, *Thymus*, *Lavandula*, *Satureja*, *Origanum*, *Mentha*, and *Rosmarinus*) and Asteraceae families (*Ditrichia*, *Santolina*, and *Tanacetum*), some extracted by two different methods (laboratory-scale hydrodistillation and pilot plant-scale steam distillation), were evaluated against *Leishmania infantum* and *Phytomonas davidi*. Additionally, the cytotoxic activity on mammalian cells of the major constituents from the most active EOs was evaluated, and their anti-*Phytomonas* and anti-*Leishmania* effects analyzed.

## 2. Materials and Methods

### 2.1. Plants and Essential Oil Extraction

The EOs studied in this work were obtained from four species of the Asteraceae family and seventeen species of the Lamiaceae family. The plant species *Tanacetum vulgare*, *Lavandula* x *intermedia* “Grosso”, *Salvia blancoana*, and *Thymus mastichina* were cultivated in distinct locations of Spain (see Appendix A for locations and voucher numbers). The plant species were identified by Dr. Daniel Gomez, IPE-CSIC, and the seeds were deposited at the CITA (Centro de Investigación y Tecnología Agroalimentaria de Aragón, Unidad de Recursos Forestales, Zaragoza, Spain) germplasm bank. Location, identification, and voucher numbers of the remaining species have been previously reported by Bailén et al. [24].

The EOs were obtained at the Research Center and Food Technology (CITA-Aragón, Spain) using two different methods, hydrodistillation (HD) and steam distillation (SD), as described by Bailén et al. [24]. Aerial plant parts were collected at the flowering stage between 2016 and 2019. The hydrodistillation was carried out in triplicate with 100 g of dried aerial plant parts and 1 L of water for 1 h in a Clevenger-type. The oils were dried over MgSO_4_, filtered and stored at 4 °C until used. Pilot plant steam distillation was carried out on the fresh biomass of the plants (60 kg total fresh plant biomass) harvested at the flowering stage. A stainless-steel pilot extraction plant equipped with a pressure reducing valve was used as described. The pressure of the work was 0.5 bar. The hydrolate (aqueous phase) was decanted from the essential oil collected in the condensation section, and filtered.

### 2.2. EO Analysis

The EOs were analyzed by gas chromatography-mass spectrometry (GC-MS) using a Shimadzu GC-2010 Plus coupled to a Shimadzu GCMS-QP2010-Ultra mass detector with an electron impact ionization source at 70 eV and a Single Quadrupole analyzer, and employing Helium as carrier gas. The samples were injected by an automatic injector (AOC-20i). Chromatography was carried out with a Teknokroma TRB-5 (95%) Dimethyl- (5%) diphenylpolysiloxane capillary column, 30 × 0.25 mm ID and 0.25 μm phase thickness. The working conditions used were: Split mode injection using 1 µL of sample with a split ratio (20:1) employing a Shimadzu AOC-20i automatic injector, injector temperature 300 °C, transfer line temperature connected to the mass spectrometer 250 °C, and ionization source temperature 220 °C. The initial column temperature was 70 °C, heating up to 290 °C at 6 °C/min, and leaving at 290 °C for 15 min. All the samples (4 g/µL) were previously dissolved in 100% dichloromethane (DCM) for injection.

The mass spectra and retention time were used to identify the compounds by comparison with those found in the Wiley database (Wiley 275 Mass Spectra Database, 2001) and NIST 17 (NIST/EPA/NIH Mass Spectral Library), while the relative area percentages of all peaks obtained in the chromatograms were used for quantification. Identification of necrodanes in *L. luisieri* was performed with our own database (CSIC), built by the injection of pure compounds isolated from this plant [25]. In addition, to confirm the identities of the constituents, the retention index of marker constituents of known EOs were used.

### 2.3. Pure Compounds

After analyzing the composition of the EOs by GC-MS, thymol, carvacrol, γ-terpinene and *p*-cymene, major components of EOs with anti-*Phytomonas* activity, were selected for further studies. Some of the major compounds that were excluded were not identified, not commercially available, or not easy to obtain or isolate. Pure compounds (monoterpenes) were obtained from commercial sources. Thymol (≥98.5%) was obtained from Sigma Aldrich (Madrid, Spain), γ-terpinene (97%) and *p*-cymene (99%) from Acros Organics (Madrid, Spain), and carvacrol (98%) from Fluka (Madrid, Spain).

### 2.4. Anti-Parasitic Activity In Vitro

Anti-*Phytomonas* (AP) and anti-*Leishmania* (AL) activity studies were performed on promastigote forms of *P. davidi* ATCC^®^ 30287™, isolated from *Euphorbia heterophylla* [26] and *L. infantum* “JPC” (MCAN/ES/98/LLM-722), kindly donated by Dr. J.M Requena from CBM-CSIC. The *P. davidi* promastigotes were cultured in LIT medium and the *L. infantum* JPC in RPMI medium, supplemented with 10% and 15% heat-inactivated fetal calf serum (FCS), respectively, at 28 °C. RPMI was also supplemented with 10 μg/mL of hemin (Acros Organics, Madrid, Spain). Parasites in the logarithmic growth phase were distributed in 96-well flat-bottom plates (95 μL of culture/well). EOs and compounds in DMSO were tested in quadruplicate at various concentrations for 24 h (*Phytomonas*) and 48 h (*Leishmania*) at several concentrations (EOs at 800, 400 and 200, 100, 50 and 25 µg/mL; and pure compounds at 100, 50, 25, 10 and 1 μg/mL). Amphotericin B was used as a reference drug. The parasite viability was analyzed by a modified MTT colorimetric assay method [27]. The percentage of AP activity and AL activity was calculated as growth inhibition using the following formula: % AP or AL = 100 − [(Ap − Ab) ÷ (Ac − Ab)] × 100; Ap is the absorbance of the tested product, Ab the absorbance of the blank, and Ac the absorbance of the control wells (not treated).

### 2.5. Cytotoxicity of Pure Compounds

African green monkey kidney cells (Vero cells) were maintained in Dulbecco’s modified Eagle’s minimal essential medium (DMEM) supplemented with 10% fetal calf serum and 1% penicillin/streptomycin (Fisher Scientific, Madrid, Spain) at 37 °C under a humidified atmosphere of 5% CO_2_/95% air.

Cells seeded in 96-well flat-bottom microplates with 100 μL medium per well (initial densities 10^4^ cells per well) were exposed for 48 h to serial dilutions (100, 75, 50, 25, 10 and 1 µg/mL) of the tested compounds in DMSO (< 1% final concentration). The cell viability was analyzed by the MTT colorimetric assay method, and the purple-colored formazan precipitate was dissolved with 100 μL of DMSO [24].

### 2.6. Statistical Analysis

The data were analyzed using STATGRAPHICS Centurion XIX (https://www.statgraphics.com, accessed on 5 September 2022). A parametric bivariate correlation analysis was performed between the main components of the EOs, present in a proportion higher than 5%, and the AL or AP activity (IC_50_).

The cells’ and promastigotes’ viabilities were tested with each compound in a dose-response experiment, to calculate their relative potency (CC_50_ or IC_50_ value). IC_50_ (μg/mL) expresses the dose of EOs or pure compounds needed to produce 50% mortality of promastigotes, while CC_50_ (μg/mL) expresses the dose of compounds necessary to produce 50% mortality of Vero cells.

The selectivity index was calculated for the AP or AL of pure compounds, using the formula SI = IC_50_/CC_50_. Compounds with an SI higher than one were considered as potential anti-*Phytomonas* or anti-*Leishmania* compounds, since they are more toxic for kinetoplastids than for mammalian cells.

## 3. Results

### 3.1. Antiprotozoal Activity of EOs from Lamiaceae and Asteraceae Plants

The activity of 23 EOs from 17 selected species (domesticated or undergoing domestication) belonging to the Asteraceae and Lamiaceae families were evaluated against the kinetoplastids *P. davidi* and *L. infantum*. All of these EOs were obtained at laboratory scale by hydrodistillation (HD). EOs from the domesticated plants cultivated in the field (a total of 17) were also extracted at a semi-industrial scale, representative of a potential commercial situation, by steam distillation (SD) (Table 1).

Eight EOs, from five *Lavandula* species, were tested against *Leishmania.* Six of them showed high antileishmanial activities at 800 µg/mL (AL > 70%), one had a moderate effect (AP: 50–70%), and one was not active (Figure 1A and Appendix A). The most effective *Lavandula* EOs were *L. luisieri* 1 (SD) and *L. luisieri* 2 (HD and SD), followed by *L. mallete* (SD) and *L. intermedia* “Abrial” (SD). The EOs with the highest anti-*Phytomonas* activity were *L. intermedia* “Super”, *L. luisieri* 1 (HD), and *L. mallete* (HD). Overall, EOs obtained by SD from *Lavandula* sp. were more active against *Leishmania* and those extracted by HD were more active against *Phytomonas*.

EOs belonging to *S. montana, M. suaveolens*, and *R. officinalis* had effects on both parasites. Differences related to the EO extraction method were observed for *R. officinalis* against both parasites (Figure 1 and Appendix A). S. *montana* EOs had the highest antiprotozoal activity against both parasites and the AP activity was maintained even at 100 µg/mL. Among the four EOs from *Salvia* spp., two were highly active against *L. infantum*, and two had no activity (Table 1). The EOs with the largest effects were *S. hybrid* (HD) and *S. officinalis* (SD). Worse results were obtained against *Phytomonas*, with three EOs having moderate effects and one showing no activity. *S. officinalis* was the *Salvia* species with the highest anti-Trypanosomatidae activity. The EOs from *Origanum* and *Thymus* spp. showed high or moderate activity against *Leishmania*. The best AL effects were obtained for *T. zygis*, and *T. vulgaris* (SD EOs). On *P. davidi, T. vulgaris* (HD) showed the highest activity at 200 µg/mL. No activity against *Phytomonas* was found for *T. mastichina*, *T. zygis* (HD), or *O. majorana*.

The EOs from the Asteraceae (*Tanacetum*, *Santolina*, and *Ditrichia)* showed moderate antiprotozoal activities, except for *T. vulgare* which showed no activity on *P. davidi* and *L. infantum.*

Overall, antiparasitic results of the tested EOs showed that *L. infantum* was more sensitive to the EOs than *P. davidi* (50% vs. 30% active EOs: IC_50_ < 500 μg/mL). The most active EOs against *P. davidi* (IC_50_ < 100 μg/mL) were *S. montana* (SD), *T. vulgaris* (HD) and *S. montana* (HD), whilst for *L. infantum* they were *M. suaveolens* (SD), *L. luisieri* 2 (SD), and *L. luisieri* 1 (SD). In general, the EOs extracted by SD performed better against *L. infantum* (100% SD EOs).

### 3.2. Chemical Composition of Essential Oils

EOs with an IC_50_ lower than 200 μg/mL were selected for further analysis (both extraction methods), adding to a total of six species and seven EOs (Table 2 and Appendix A; Appendix A). Only components with a proportion higher than 5% were considered.

Camphor, trans-α-necrodyl acetate, and fenchone were predominant in *L. Luisieri* (1 and 2), with fenchone being more effectively extracted by HD and trans-α-necrodyl acetate by SD. For *M. suaveolens*, the main components were piperitenone and piperitenone oxide, which accounted for more than 60% of the total composition. Piperitenone was more abundant in the EO obtained by HD and piperitenone oxide in the EO obtained by SD. Among the components of *S. hybrid* HD, 1,8 cineol, camphor, and trans-bornyl acetate appeared in higher proportions.

In the chemical composition of *S. montana*, thymol, *p*-cymene, and carvacrol stand out, with thymol and *p*-cymene being more abundant when extracted by HD and carvacrol when the extraction was done by SD. Thymol and *p*-cymene were also the main components of *T. vulgaris* and *T*. *zygis*. Linalyl acetate was also an abundant component, but only for *T. zygis* SD EO.

Overall, HD extraction increased borneol and 1,8 cineol, whilst SD extraction favored linalyl acetate, β-caryophyllene, and trans-α-necrodyl acetate.

A parametric bivariate correlation analysis was performed between the main components of the EOs and the AL and AP activities. There were two significant correlations for AL and three for AP (*p* < 0.05) (Table 3). Among the compounds with significant correlations, piperitenone oxide was directly associated with the AL activity (negative correlation between compound abundance and IC_50_ of the EO), whereas *p*-cymene, thymol, and carvacrol were directly associated with the AP activity of the EOs. Only α-terpineol was inversely associated with AL activity (positive correlation between compound abundance and IC_50_ of the EO).

### 3.3. Anti-Phytomonas and Cytotoxic Activity of Pure Compounds

A total of four compounds (Figure 2), selected from the chemical composition of the most active EOs, were tested on *L. infantum* and *P. davidi*, and also their cytotoxic effects on Vero cells were evaluated (Table 4). Amphotericin B was used as a reference drug. Data previously reported from these compounds on *Leishmania* sp. have been included in Table 4. Only compounds **1** (thymol) and **2** (carvacrol) showed AP activities with moderate effects, with **1** (thymol) being the compound with the highest AP activity (IC_50_: 45 µg/mL). The highest SI was observed for thymol (SI: 2.22). Compounds **1** (thymol) and **2** (carvacrol) were also the terpenes with the highest selectivity indexes (SI: 10.20 and 13.85 respectively) on *L. infantum. p*-Cymene (**4**) was the only terpene previously tested on other *Leishmania* species (*L. chagasi)* [28]. None of the tested compounds were cytotoxic against Vero cells.

## 4. Discussion

Lamiaceae is an important family, with a variety of aromatic and medicinal genera such as *Rosmarinus*, *Origanum*, *Thymus*, *Lavandula*, *Mentha*, and *Satureja*, among many others [31]. The most active EOs found in our study for both kinetoplastids, *L. infantum* and *P. davidi*, belong to this family, highlighting EOs from *M. suaveolens* and *S. montana*. The Asteraceae plant family has emerged as a new source of trypanocidal and leishmanicidal compounds [32]. However, the Asteraceae EOs tested in this study only had moderate effects on both targets.

Overall, the results of the tested EOs in our study showed that *L. infantum* was more sensitive to the EOs than *P. davidi*, probably because *Phytomonas* is a plant parasite, and thus it is more adapted to the chemical defensive components of plants. These results agree with those found by Sainz et al. when comparing the activity of various EOs on *T. cruzi* and *P. davidi* [27]. The activity of EOs on *Phytomonas* sp. have been less studied than the leishmanicidal effects. Even so, EOs from other plant families have been found to have anti-*Phytomonas* activity. EOs from *Varronia curassavica* genotypes have antiprotozoal activity against *Phytomonas serpens* causing alterations in the permeabilization of the cytoplasmic membrane [33]. *Lantana camara* EOs, extracted at different harvesting times, displayed trypanocidal activity on *P. serpens* [34], and EOs from *Hyssopus officinalis* showed antiprotozoal activity on *T. cruzi* and *P. davidi* [35]. However, this is the first report on the phytomonacidal effects of the EOs studied here.

*M. suaveolens* is an aromatic species usually employed in traditional medicine, with reported cytotoxic, antioxidant, anti-inflammatory, antifungal, antiviral, and insecticidal properties [36]. Our results indicate that *M. suaveolens* EO had the highest antileishmanial effects of all the tested EOs, as well as moderate effects on *Phytomonas*. This EO contained high proportions of piperitenone oxide and piperitenone. Piperitenone oxide could be involved in the antileishmanial activity, as it has been previously been reported to have trypanocidal, insecticidal, and schistosomicidal effects [37,38,39]. There are also other species of the genus *Mentha* with antileishmanial properties on promastigotes of various species of *Leishmania*. EOs from *M. australis* and *M. microphylla* have antileishmanial effects on *L. major* [40], *M. pulegium* on *L. major*, *L. infantum* and *L. tropica* [41], *M. x piperita* on *L. infantum* and *L. donovani* [42,43], and *M. cervina* on *L. infantum* [42,44]. However, none of the mentioned EOs had piperitenone oxide as a major component.

Another plant species with strong anti-kinetoplastid activity in our study was *S. montana*. The genus *Satureja* comprises numerous species of shrubs or aromatic herbs with acaricidal, insecticidal, and antiparasitic properties [45]. *Satureja montana* EO has interesting properties against bacteria, fungi, viruses, helminths, insects, and protozoa [24,46,47,48,49]. EOs from *Satureja khuzestanica*, *Satureja bakhtiarica*, and *Satureja punctata* demonstrated antileishmanial effects on *L. major*, *L. donovani*, and *L. aethiopica* previously [50,51,52]. *S. montana* EO (HD and SD) was active against both parasites, with *P. davidi* being more sensitive. Thymol and carvacrol, which correlated with AP activity, were the main components of the *Satureja* EO.

EOs from two populations of *L. luisieri* (1 and 2) had the strongest effects among the tested lavandulas on *L. infantum*. Essential oils from plants of the *Lavandula* genus have reported acaricidal, antibacterial, antifungal, antioxidant, and anti-parasitic effects [53]. EOs from *L. angustifolia, L. stoechas*, *L. viridis*, and *L. luisieri* had reported effects on *L. major*, *L. infantum*, and *L. tropica* [42,53,54,55]. However, the effect found here was weaker than that observed by Machado et al. on *L. infantum* promastigotes [53]. Trans-α-necrodyl acetate is one of the major components of *L. luisieri* EOs. Therefore, necrodane derivatives should be further studied to verify their antileishmanial properties. *L. luisieri* EOs can produce leishmanicidal effects through different mechanisms, but mainly through unleashed apoptosis, with phosphatidylserine externalization, mitochondrial membrane potential loss, and G0/G1 phase cell cycle arrest being the most remarkable aspects involved in apoptosis [53]. On *P. davidi*, the studied *Lavandula* EOs only had moderate effects, with *L. intermedia* “Super” being the most active.

*T. vulgaris* HD EO, showing a remarkable anti-*Phytomonas* activity, had the highest content of thymol (48%) of all the tested EOs. As mentioned before, thymol correlated with AP activity. EOs from *T. zygis*, *Thymus capitelatus*, and *T. mastichina* have been reported to have antileishmanial effects on *L. infantum* promastigotes [42,56].

Among the genus *Salvia*, the EOs from *S. hybrid* and *S. officinalis* showed important antileishmanial effects, while *S. blancoana* EOs had moderate action against *P. davidi*. EOs from various species of *Salvia* have been previously tested on *Phytomonas* and *Leishmania*, with moderate effects [17,22].

In this study, the EO (HD and SD) from *R. officinalis* showed a moderate effect on *L. infantum.* Previous reports have showed strong antileishmanial effects of *R. officinalis* EOs on *L. major*, *L. tropica*, and *L infantum* [41,54]. Variations in the chemical composition of the EOs could explain these differences.

*Origanum majorana* is a shrub found in Asia and in the Mediterranean area, with reported antibacterial, antifungal, antiparasitic, anthelmintic, and antiviral activities [57,58,59]. EOs from *O. majorana* and *O. dubium* have been reported to have antimalarial effects on mice, reducing the parasitemia and increasing their life span [57]. Also, EOs from *O. majorana*, *O. virens*, and *O. vulgare* have been reported to have anti-*Phytomonas* effects on *P. davidi*, with *O. virens* having the strongest effect [22]. Other antileishmanial effects were reported from *O. virens* EO on *L. infantum* [42], and *O. vulgare* EO on *L. amazonensis*, *L. panamensis*, and *L. braziliensis* [60,61].

In our study, thymol (**1**) and carvacrol (**2**) correlated with AP activity, and were active when tested on *P. davidi*, while *p*-cymene was not. Thymol and carvacrol are *p*-cymene derivatives, and therefore their amounts in EOs are usually correlated, explaining the correlation of this compound with AP effects. Compounds **1** and **2** are common in EOs with activity against promastigotes and amastigotes of *L. infantum* [29]. Thymol has been reported to have anti-*Phytomonas* [22] and leishmanicidal properties on promastigotes of *L. infantum* [29], promastigotes and amastigotes of *L. infantum,* and amastigotes of *L. donovani* [62]. The anti-*Leishmania* and anti-*Phytomonas* activities of carvacrol and thymol depend on the presence of the phenolic hydroxyl group, as observed before by Silva et al. [63]. The lack of the phenolic hydroxyl group in the precursor *p*-cymene is associated with the absence of antiprotozoal effects. Also, thymol has been used as a starting compound to obtain derivatives with stronger antileishmanial effects [64].

The biological properties of EOs can be determined by their major compounds, but minor compounds modulate these effects. Synergistic effects between EO components have been observed before. A mixture 1:4 of lupenone and β-caryophyllene oxide presented better antileishmanial activity and lower cytotoxicity than β-caryophyllene oxide alone [65]. Also, the combination between ascaridole and carvacrol produced synergistic effects on *L. amazonensis* [66].

## 5. Conclusions

EOs with better anti-kinetoplastid activity were *S. montana*, *T. vulgaris*, *M. suaveolens*, and *L. luisieri*. They are good sources of thymol, carvacrol, trans-α-necrodyl acetate, and piperitenone oxide. Further studies should be performed with trans-α-necrodyl acetate and piperitenone oxide to corroborate their potential activity against *L. infantum* and *P. davidi.* Thymol and carvacrol were the best anti-kinetoplastid compounds of this study. The synthesis of new compounds, using carvacrol and thymol as starting compounds, could be a strategy for the search for new anti-trypanosomatid compounds as alternatives to the current treatments, due to their anti-*Phytomonas* and antileishmanial effects.

*Phytomonas* is a plant parasite that can cause important economic losses and that lacks an effective treatment. Our study identifies potential treatments against this pathogen, and extraction methods which potentiate the concentration of specific components. The potential use of EOs, and their main components, as new alternatives for the treatment of animal trypanosomatid diseases, including leishmaniasis, could lie in their use as alternative treatments or in combination with approved treatments, to increase the efficiency and diminish the toxic effects of the current therapeutic protocols.

## Figures and Tables

**Figure 1 molecules-28-01467-f001:**
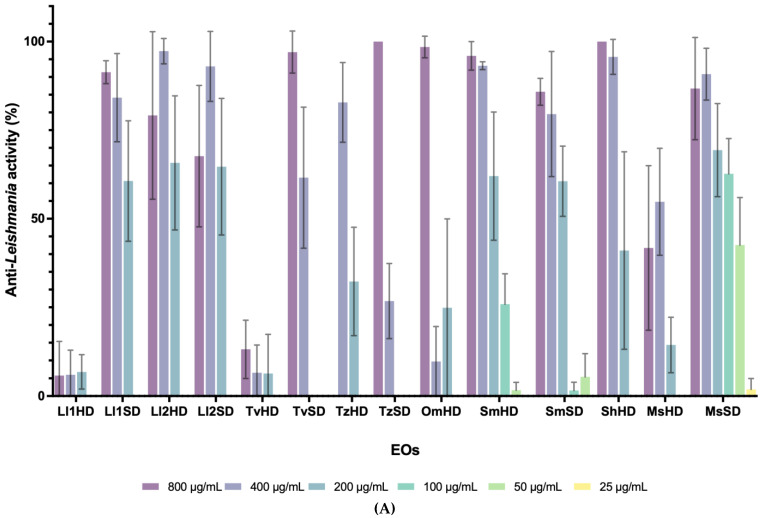
Percentage of antiprotozoal activity of EOs with IC_50_ < 200 µg/mL obtained by hydrodistillation (HD) and steam distillation (SD) from *L. luisieri* 1 and 2 (Ll1 and Ll2), *T. vulgaris* (Tv)*, T. zygis* (Tz), *O. majorana* (Om), *S. hybrid* (So), *S. montana* (Sm), and *M. suaveolens* (Ms). (**A**) Anti-*Leishmania* activity. (**B**) Anti-*Phytomonas* activity of EOs.

**Figure 2 molecules-28-01467-f002:**
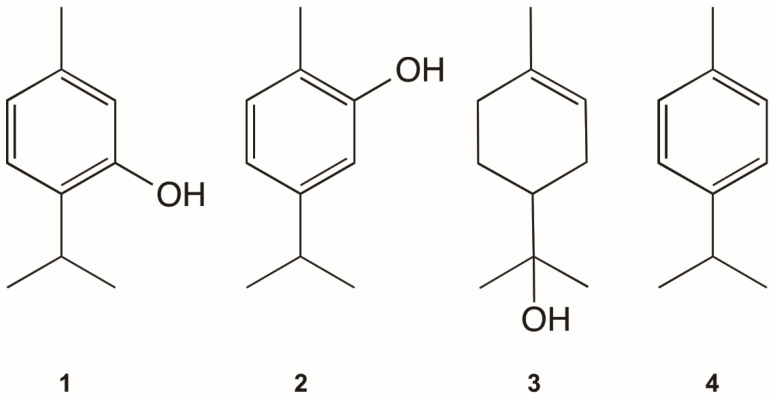
Chemical structures of the tested monoterpenes.

**Table 1 molecules-28-01467-t001:** Effects of the tested EOs from Lamiaceae and Asteraceae species on *L. infantum* and *P. davidi* (IC_50_) ^a^.

Family	Genus	Species	Extraction Method	*L. infantum* (IC_50_)	*P. davidi* (IC_50_)
Asteraceae	*Tanacetum*	*Tanacetum vulgare*	HD	325.5 (300.0, 353.1)	>800
	*Santolina*	*Santolina chamaecyparissus*	HD	451.2 (391.4–520.2)	599.2 (580.7–618.3)
	*Ditrichia*	*Ditrichia graveolens*	HD	350.8 (303.4–405.4)	341.5 (288.8–403.8)
			SD	306.8 (246.7–381.5)	>800
Lamiaceae	*Lavandula*	*Lavandula lanata*	HD	>800	>800
		*Lavandula luisieri* 1	HD	>800	572.5 (557.5–587.9)
			SD	<200	439.5 (394.8–489.3)
		*Lavandula luisieri* 2	HD	247.4 (227.7–268.8)	573.9 (485.6–678.2)
			SD	74.3 (61.6–89.5)	>800
		*Lavandula angustifolia*	HD	461.2 (402.2–528.8)	>800
			SD	512.2 (477.1–550.0)	>800
		*Lavandula x intermedia* “Abrial”	HD	634.0 (528.5–760.7)	>800
			SD	319.1 (294.8–345.3)	>800
		*Lavandula x intermedia* “Super”	HD	466.5 (404.2–538.3)	343.8 (259.3–455.8)
			SD	640.7 (603.6–680.1)	>800
		*Lavandula x intermedia* “Grosso”	HD	>800	496.6 (417.9–590.1)
			SD	633.8 (595.0–675.0)	>800
		*Lavandula mallete*	HD	397.9 (362.4–436.9	436.5 (367.4–518.6)
			SD	293.2 (233.8–367.6)	>800
	*Origanum*	*Origanum majorana*	HD	547.9 (521.4–575.7)	>800
		*Origanum virens*	HD	781.2 (742.7–821.7)	507.8 (486.8–529.7)
			SD	742.6 (664.7–829.6)	521.9 (499.0–545.8)
	*Rosmarinus*	*Rosmarinus officinalis*	HD	718.9 (667.1–774.7)	599.6 (495.4–725.7)
			SD	367.1 (307.8–437.7)	446.5 (321.7–619.7)
	*Satureja*	*Satureja montana*	HD	170.9 (154.0–189.6)	83.1 (67.2–102.7)
			SD	194.1 (176.7–213.3))	49.7 (46.0–53.7)
	*Mentha*	*Mentha suaveolens*	HD	117.9 (101.3–137.4)	431.6 (404.5–460.6)
			SD	88.2 (73.2–106.4)	396.0 (374.0–419.3)
	*Salvia*	*Salvia officinalis*	HD	680.8 (621.8–745.4)	>800
			SD	<200	329.3 (280.6–386.4)
		*Salvia hybrid*	HD	184.7 (154.8–220.3)	406,3 (387.2–426,3)
		*Salvia blancoana*	HD	>800	514.2 (474.1–557.7)
		*Salvia sclarea*	HD	>800	>800
	*Thymus*	*Thymus mastichina*	HD	743.7 (682.4–810.5)	>800
		*Thymus vulgaris*	HD	>800	43.3 (33.5–56.0)
			SD	375.6 (331.8–425.1)	426.4 (369.9–491.4)
		*Thymus zygis*	HD	141.8 (99.1–202.8)	>800
			SD	433.0 (394.9–474.7)	182.3 (155.7–213.4)

^a^ IC50 (μg/mL) = concentration needed to produce 50% promastigote mortality. SD: steam distillation. HD: hydrodistillation. Light grey: IC_50_ < 200 μg/mL; Dark grey: IC_50_: 200–500 μg/mL.

**Table 2 molecules-28-01467-t002:** Main components of the EOs from the most active species (abundance ≥ 5%).

Plant Species	EM	Compounds (% Relative Abundance)
*M. suaveolens*	HD	piperitenone (53%), piperitenone oxide (23%), limonene (6%)
	SD	piperitenone oxide (37%), piperitenone (21%), limonene (7%), germacrene D (7%), β -caryophyllene (6%)
*L. luisieri* 1 *	HD	fenchone (20%), camphor (13%), trans-α-necrodyl acetate (12%), lavandulyl acetate (6%), α-pinene (6%), 1,8-cineole (5%)
	SD	camphor (35%), trans-α-necrodyl acetate (19%), lavandulyl acetate (6%), α-pinene (5%)
*L. luisieri* 2 *	HD	camphor (49%), trans-α-necrodyl acetate (13%), lavandulol (6%)
	SD	trans-α-necrodyl acetate (18%), lavandulol (8%), germacrene D (8%), camphor (5%)
*S. hybrid* *	HD	1,8-cineole (21%), camphor (14%), trans-bornyl acetate (13%), β-pinene (11%), camphene (7%)
*S. montana* *	HD	carvacrol (33%), *p*-cymene (18%), thymol (17%), γ-terpinene (12%)
	SD	carvacrol (41%), *p*-cymene (12%), γ-terpinene (12%), thymol (7%), β-caryophyllene (6%)
*T. vulgaris* *	HD	thymol (43%), *p*-cymene (22%), γ-terpinene (6%)
	SD	carvacrol (41%), *p*-cymene (31%), thymol (28%), β-caryophyllene (6%)
*T. zygis* *	HD	thymol (39%), *p*-cymene (18%), γ-terpinene (9%), linalool (5%), borneol (6%), carvacrol (5%)
	SD	thymol (21%), linalyl acetate (18%), linalool (12%), *p*-cymene (8%), carvacrol (5%), α-bisabolol (5%)

EM: extraction method; HD: EOs obtained by hydrodistillation; SD: EOs obtained by steam distillation; *: previously reported [25].

**Table 3 molecules-28-01467-t003:** Pearson correlation coefficients between main components of EOs and AL and AP activities.

Compound	AL Activity		AP Activity	
	PCC	*p*	PCC	*p*
*p*-cymene	−0.048	0.773	−0.441	0.005
α-terpineol	0.408	0.010	0.247	0.129
piperitenone oxide	−0.338	0.035	−0.146	0.377
thymol	−0029	0.859	−0.396	0.013
carvacrol	−0242	0.138	−0.517	0.001

PCC: Pearson correlation coefficient.

**Table 4 molecules-28-01467-t004:** Effects of the tested compounds on *P. davidi, L. infantum*, and Vero cells (IC_50_ and CC_50_ respectively).

Compound	Vero Cells [25]	*P. davidi*		*L. infantum*	
	CC_50_ (µg/mL) ^a^	IC_50_ (µg/mL) ^b^	SI ^c^	IC_50_ (µg/mL) ^b^	SI ^c^
thymol (**1**)	≈100	45.0 (38.4–52.7)	2.22	7.22 (6.22–8.62) [29]	13.85
carvacrol (**2**)	>100	79.1 (71.9–86.9)	1.26	9.8 (8.51–11.7) [29]	10.20
α-terpineol (**3**)	>100	>100	1	nd	-
*p*-cymene (**4**)	>100	>100	1	>100 [28]	0.67
amphotericin B	-	0.2 (0.1–0.2)	-	0.01 [30]	10.000

^a^ CC50 (μg/mL) = concentration needed to produce 50% Vero cell mortality; ^b^ IC50 (μg/mL) = concentration needed to produce 50% trophozoite mortality, ^c^ SI: Selectivity index.; nd: not determined.

## Data Availability

Not applicable.

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
