# Peer review of "Anti-Trypanosomatidae Activity of Essential Oils and Their Main Components from Selected Medicinal Plants"

_molecules, 2023, doi:10.3390/molecules28031467_

Round 1
Reviewer 1 Report
I would like to congratulate the authors on the high quality of the papers. The paper reads well and is very easy to follow. I would recmomend the Authors should provide a graphical abstract to increase appeal.
As a suggestion for future works I propose that associations of EO, or EO components, with standard pharmacological treatment should be assessed in order to verify the possibility of a synergic action or to identify interactions.
Author Response
Dear reviewer,
We have included a graphical abstract and will follow your suggestion about synergism for future works.

Reviewer 2 Report
I have carefully read the manuscript by Bailén et al in which the anti-Trypanosomatidae activity of essential oils and their main components from selected medicinal plants has been reported. In my opinion, the article is well written and has novelty and the experiments are well conducted.
I suggest to check just some typos or few minor grammar mistakes before possible publication in "Molecules" journal.
Moreover, for compounds purchased for in vitro experiments the authors should indicate the grade of purity (in paragraph 2.3)
Author Response
Dear reviewer,
We have included the grade of purity of the purchased compounds and checked carefully for typos and grammar mistakes along the manuscript.
